# How Does Selenium Intake Differ among Children (1–3 Years) on Vegetarian, Vegan, and Omnivorous Diets? Results of the VeChi Diet Study

**DOI:** 10.3390/nu15010034

**Published:** 2022-12-21

**Authors:** Stine Weder, Esther H. Zerback, Sina M. Wagener, Christian Koeder, Morwenna Fischer, Ute Alexy, Markus Keller

**Affiliations:** 1Research Institute for Plant-Based Nutrition (IFPE), 35444 Biebertal, Germany; 2Faculty of Human Resources, Health & Social Work, University of Applied Sciences (FHM), 50674 Cologne, Germany; 3IEL-Nutritional Epidemiology, DONALD Study, University of Bonn, 44225 Dortmund, Germany

**Keywords:** child nutrition, vegans, vegetarians, veganism, mixed diet, plant-based diet, toddlers, critical nutrients, Brazil nuts

## Abstract

In regions with low selenium soil concentrations, selenium can be considered a critical nutrient for vegetarians and vegans. While the number of vegetarians and vegans is increasing in many countries, a large research gap remains in this field. For example, to date, no study seems to have assessed selenium intake in vegetarian and vegan children. Therefore, the selenium intake of 1- to 3-year-old vegetarian, vegan, and omnivorous children who participated in the cross-sectional VeChi Diet study was determined. Selenium intake was assessed based on 3-day food diaries (not including supplements) and food selenium concentrations provided by the European Food Safety Authority (EFSA). Between-group differences were assessed with analysis of covariance (ANCOVA). The median daily selenium intake was 17 µg, 19 µg, and 22 µg in vegetarian, vegan, and omnivorous children, respectively. However, only the difference between the vegan and omnivorous children was statistically significant. On average, all three groups met the harmonized average requirement (H-AR) for selenium of 17 µg/day. Nevertheless, the hypothesis that vegetarian and vegan children generally consume less selenium than omnivorous children could be confirmed, and 39% of vegetarians, 36% of vegans, and 16% of omnivores fell below the adequate intake for selenium (provided by EFSA) of 15 µg/day.

## 1. Introduction

The number of vegetarians and vegans in Germany is increasing, and many vegetarian and vegan parents feed their children accordingly [1]. If the diet is not adequately planned, vegetarian and vegan diets may result in an inadequate intake of certain nutrients [2]. This is particularly the case with young children, as they have increased energy and nutrient requirements for the processes of growth and development. Thus, it is necessary to ensure an adequate intake of potentially critical nutrients such as vitamin B12, protein, long-chain omega-3 fatty acids, vitamin B2 (riboflavin), calcium, zinc, iodine, iron, and selenium [3].

In foods of animal origin, such as offal and fish (as well as other meats and seafood), selenium is generally found in intermediate to high concentrations. In contrast, selenium concentrations in plant foods are highly variable and particularly depend on soil selenium levels, which are typically low in Central and Northern Europe [4,5]. Thus, the avoidance of animal foods can result in a lower selenium intake [6].

However, to date, no studies have assessed the selenium intake of vegan children [7]. One of the most recent studies in the field of vegetarian and vegan child nutrition is the VeChi Diet study from Germany, whose participants were 1- to 3-year-old vegetarian, vegan, and omnivorous children [1,8]. In the initial evaluations of this study (results of which have been published [1,8]), the selenium concentrations of the foods consumed were not available. Therefore, the present short communication presents the results regarding selenium intake of the VeChi Diet participants, i.e., vegetarian, vegan, and omnivorous children, aged 1 to 3 years. Thus, the results of the present study will make a valuable contribution to assessing the adequacy of vegetarian and especially vegan dietary patterns for children.

## 2. Materials and Methods

### 2.1. Study Design and Participants

The VeChi Diet study is a cross-sectional study with the aim of investigating anthropometric parameters and dietary intake among vegetarian, vegan, and omnivorous children. The study was conducted from 2016 to 2018 and included a total of 430 healthy children aged 1–3 years (living in Germany).

The detailed study design and results regarding anthropometric parameters and nutrient intake have been published [1,8].

As some studies have shown a lower selenium intake in vegetarian and vegan adults compared to omnivores [2], the hypothesis of the present study was that selenium intake would be lower among vegetarian and vegan children.

### 2.2. Data Assessment

In the present study, the term “vegetarian” was defined as lacto-ovo-vegetarian (including dairy and eggs, excluding fish and other seafood). “Vegan” was defined as excluding all animal-derived foods from the diet (for further details, see [1]). Food intake was assessed with 3-day weighed dietary records. To estimate breast milk intake, the number of breast milk meals reported in the present study was multiplied by the age-specific median volume of breast milk per breast milk meal reported in the DONALD study (Germany) (as reported previously [1,9]). Energy and standard nutrient intake were calculated as individual means over three days using the food composition database LEBTAB [1,8]. As LEBTAB does not contain data on selenium, the foods contained in the dietary records were linked to the selenium food concentrations provided by the European Food Safety Authority (EFSA) food composition database [10,11]. To classify and describe foods, FoodEx2 codes were necessary, which are part of a standardized system developed by EFSA. The FoodEx2 system consists of descriptions of many individual foods, which are grouped into 21 food groups in hierarchical parent-child relationships. To link the consumption data with the selenium concentrations, all foods consumed in this study also had to be assigned FoodEx2 codes. 

For the compilation of this database, 14 European countries were included, but selenium concentrations were only available from nine countries [10]. Of these, four countries (Denmark, France, the Netherlands, and Sweden) were chosen for this analysis because, like Germany, they have intermediate to low soil selenium concentrations compared to other European countries [12]. In total, for these four countries, 5852 food entries and the corresponding selenium concentrations were available. If there were multiple entries for a food, the mean of these entries was used, unless one value deviated significantly (defined as an outlier in terms of selenium content: >3rd quartile + 1.5*interquartile range [IQR] or <1st quartile—1.5*IQR). These outliers were excluded from the calculation.

The consumption of Brazil nuts was considered particularly important for the calculation of selenium intake because of the high selenium content of these nuts and the ability of this tree to accumulate selenium in high amounts [13]. The EFSA database does not currently contain selenium concentrations for Brazil nuts. Therefore, the average selenium concentration (3250 µg of Se per 100 g of Brazil nuts) from a study that examined Brazil nuts of Brazilian origin was used [13]. Selenium intake from supplements was not available in the VeChi Diet Study and was therefore not included in dietary selenium intake. If any of the children categorized as vegetarian or vegan also consumed small amounts of animal-source foods, selenium intake from these animal-source foods was also included.

### 2.3. Data Analysis and Statistics

All statistical analyses were conducted using IBM SPSS Statistics (version 28.0; Armonk, NY, USA) or SAS (version 9.4, Cary, NC, USA). Daily selenium intake (µg/day) was determined as the individual mean of a 3-day dietary record. In a sensitivity analysis, average selenium intake excluding selenium intake from Brazil nuts was calculated to investigate whether total selenium intake was strongly influenced by selenium intake from these particular nuts. Furthermore, selenium intake per 1000 kcal was calculated to standardize energy intake. 

To estimate each group’s risk of selenium deficiency, the percentage of selenium intake compared to the harmonized average requirement (%H-AR [14]) was calculated. The H-AR for selenium for 1- to 3-year-olds is 17 µg/day [14] and is based on the Estimated Average Requirement (EAR) given by the Institute of Medicine (IOM, Washington, DC USA) in the year 2000 [15]. In addition, the intake was compared with the harmonized tolerable upper level of intake (H-UL), proposed by Allen et al. (2020) [14]. Moreover, selenium intake by food group was calculated to determine the main contributors to selenium intake.

Between-group differences in selenium intake were assessed by analysis of covariance (ANCOVA). The covariates age (years), sex (female; male), total energy intake (kcal/day), physical activity (less active; active; very active), socioeconomic status (low; middle; upper), urbanicity (rural/small-size; urban/medium-size; urban/metropolitan), season (spring; summer; autumn; winter), and weight-to-height Z-scores (quintiles) were adjusted for. In the basic model, the covariates age and sex were included. Interaction effects were tested for each additional covariate in the basic model. In the fully adjusted model, covariates with a *p*-value of ≤0.1 and/or a partial eta squared (η^2^) of 0.06 were included (backward method). If the assumptions of ANCOVA were violated, variables were log-transformed (log[x]). A *p*-value of ≤0.001 was considered statistically significant because of the high risk of type 1 errors due to the large number of statistical tests. An η^2^ ≥ 0.01 was interpreted as a small effect size, η^2^ ≥ 0.06 as a medium effect size, and η^2^ ≥ 0.14 as a large effect size.

## 3. Results

53% of the vegetarian, 58% of the vegan, and 52% of the omnivorous children were <2 years old. 29% of the vegetarian, 27% of the vegan, and 27% of the omnivorous children were ≥2 and <3 years old, while 18% of the vegetarian, 14% of the vegan, and 21% of the omnivorous children were already 3 years old. More detailed characteristics of the participants have been published previously [1].

Unadjusted median selenium intake was highest in the omnivorous group (22 µg/day), intermediate in the vegan group (19 µg/day), and lowest in the vegetarian group (17 µg/day; Table 1). The unadjusted energy-standardized selenium intake of omnivorous children was also the highest. However, after adjustment for age, sex, weight-to-height Z-scores, season, socio-economic status, and energy intake, ANCOVA revealed a significant difference in selenium intake only between the vegan and omnivorous groups, with a small effect size (the adjusted selenium intake is shown in the Appendix A). Nevertheless, on average, the H-AR of 17 µg/day was (almost) achieved by all diet groups (vegetarian: 99%; vegan: 110%; omnivorous: 127%). On an individual basis, 52% of vegetarians, 47% of vegans, and 28% of omnivores had a selenium intake below the H-AR (17 µg/day), and 39% of vegetarians, 36% of vegans, and 16% of omnivores had a selenium intake of <15 µg/day (15 µg/day is the adequate intake for selenium for 1- to 3-year-olds given by EFSA [10]). Regarding excessive selenium intake, the H-UL of 60 µg/day was not reached by any of the omnivorous children, whereas the H-UL was exceeded by two vegetarian children (2% of the children; exceeding the H-UL by 238% and 287%, respectively) and nine vegan children (9% of the children; exceeding the H-UL by 110–775%). According to the questionnaire, 2.4% of vegetarians, 11.5% of vegans, and 0.6% of omnivores took a selenium (or selenium-containing) supplement (with the amounts of selenium in these not known).

Furthermore, the sensitivity analysis excluding the intake of Brazil nuts in the fully adjusted model also revealed a significant difference in selenium intake between the vegan and omnivorous groups with a small effect size: vegans consumed less selenium (18 µg/day) than omnivores (22 µg/day). Similarly, the median selenium intake per 1000 kcal in the final adjusted ANCOVA model differed significantly only between vegan and omnivorous children. 

Regarding the impact of different food groups on selenium intake (Figure 1), grains turned out to be the main contributor to dietary selenium intake for all groups (vegetarian: 32%; vegan: 31%; omnivorous: 22%). Further important selenium sources were legumes (11%) and dairy (11%) for vegetarians, and nuts (including Brazil nuts; 19%) and legumes (17%) for vegans. Important contributors to selenium intake for omnivores were dairy (18%) and meat (13%; Figure 1).

It should be noted that 7 of the 139 vegan children (5%) consumed some lacto-ovo-vegetarian food when in day care, and 3 of the 127 vegetarian children (2%) also consumed some non-vegetarian food when in day care. However, the contribution of these animal-source foods to the respective selenium intakes of vegan and vegetarian children was minimal (<0.5%; Figure 1).

## 4. Discussion

In the present article, the selenium intake of vegetarian, vegan, and omnivorous children (1–3 years old) was investigated for the first time. The median selenium intake was lowest in the vegetarian group, highest in the omnivorous group, and intermediate in the vegan group. These results are in accordance with the results of the EPIC-Oxford study, which included >30,000 adults (omnivores, fish eaters, vegetarians, and vegans) in the United Kingdom and documented a lower selenium intake in vegetarians and vegans [16]. Similarly, other studies with adults [17,18,19,20,21], including one with young adults [22], have shown a lower selenium intake and/or status with vegetarian and/or vegan diets compared to omnivorous diets. However, no differences between the different diet groups (vegetarian, vegan, and omnivorous) were found in a study by Schüpbach et al. (2017) from Switzerland [23].

A possible explanation for a lower selenium intake in vegetarians may simply be that they do not consume meat or fish and (like in the present study) fewer dairy products compared to omnivores (data not shown), whereas omnivores obtained ~50% of their dietary selenium from animal-source foods (meat, dairy products, eggs, and fish), which tend to be rich in selenium (Figure 1). Thus, the expectation that selenium intake would be higher in omnivorous children due to their intake of animal-source foods was confirmed. In contrast, compared to the vegetarian children, the vegan children appear to have consumed more selenium via Brazil nuts and other nuts, as well as legumes.

The present study showed an unadjusted median selenium intake in the range of 17–22 µg/day. This result appears plausible, as 1- to 2-year-old children in the German VELS Study (2003) had a comparable median selenium intake (~17–18 µg/day) [24].

In the fully adjusted model, significant differences in median selenium intake were only found between vegan and omnivorous children, with the vegans having a lower intake, and this was independent of whether Brazil nuts were included or excluded. However, the effect size of group differences was small, and all three dietary groups achieved or exceeded the H-AR (17 µg/day) on average. Therefore, the clinical relevance of the observed lower selenium intake in vegetarian and vegan children (compared to omnivorous children) is uncertain, and the risk of inadequate selenium intake among young vegetarian and vegan children in Germany may generally be low. However, on an individual basis, 39% of vegetarians, 36% of vegans, and 16% of omnivores consumed less than 15 µg of selenium per day (the adequate intake for selenium given by EFSA for 1- to 3-year-olds [10]). It should be taken into account that in the German-speaking countries (Germany, Austria, and Switzerland) [25], the dietary reference value for selenium for 1- to 4-year-old children is 15 µg/day, while in the United States, the dietary reference value for 1- to 3-year-olds is 20 µg/day [15]. Thus, the majority of the children in the present study did not reach the selenium intake recommended in the United States.

However, it should also be taken into account that, according to the questionnaires, ~2% of vegetarians, ~12% of vegans, and ~1% of omnivores took a selenium supplement, and these selenium amounts (intake from supplements) were not included in our results (as the amounts consumed via supplements were not assessed). Thus, the actual total selenium intake, especially for vegans, could be higher than the values reported in the present study.

Nine vegan and two vegetarian children exceeded the H-UL of 60 µg/day (by 110–775%). These excessive intakes were mainly caused by the consumption of Brazil nuts. Already, one single Brazil nut may exceed the H-UL for selenium, but the variability of the selenium content of Brazil nuts is high, and selenium concentrations from the literature may not reflect real-world conditions [26]. Caregivers of vegetarian and vegan children should be advised to avoid a chronically high intake of Brazil nuts for their children. However, to our knowledge, no case report of selenium toxicity from Brazil nuts in humans has ever been reported [27]. Future studies should assess the selenium status of vegetarian and vegan children and should also consider Brazil nut intake (as well as selenium intake from supplements). 

In terms of selenium concentrations in Brazil nuts, it should be noted that, globally, Brazil and Bolivia are the main producers of these nuts and that the two countries have a similar market share [28]. Furthermore, a large percentage of Brazil nuts available to consumers in Europe seem to be of Bolivian origin (own data from a market survey, not published). At the same time, there appears to be a paucity of studies that have assessed the selenium content of Brazil nuts from Bolivia [29,30], and the two studies that could be identified found selenium concentrations of 160 µg/100 g (Brazil nuts bought in the United States) [29] and 510 µg/100 g (bought at a supermarket in Belgium) [30], respectively. In addition, these two studies observed a higher selenium content in Brazil nuts from Brazil (~2 to 10 times higher) compared to those from Bolivia [29,30]. In the present study, a relatively high selenium content (3250 µg/100 g) of Brazil nuts was assumed [13]. Therefore, the potential risk of ingesting excessive amounts of selenium by eating Brazil nuts may be considerably lower, under real-life conditions in Europe, than the results of the present study suggest. The FoodData Central database of the United States Department of Agriculture (USDA) lists a selenium content of Brazil nuts in the range of 140–2740 µg/100 g [31]. Furthermore, as a high within-country variability of selenium concentrations in Brazil nuts appears likely, relying on the data currently available in the literature is highly speculative. Thus, future studies should assess the selenium content of Brazil nuts available in supermarkets and other retailers in Europe (and other regions with low selenium soil concentrations).

The results of the present study regarding the contribution of different food groups to selenium intake confirm the results of previous studies [24,32] which showed that, in vegetarian and vegan diets, grains, nuts, and legumes strongly contribute to selenium intake.

### Strengths and Limitations

Strengths and limitations of the general study design have been described elsewhere [1]. In brief, the cross-sectional nature of the study does not allow conclusions about the long-term selenium intake of the participants. Furthermore, selenium status was not determined in the present study, which strongly limits the interpretation of the results. Furthermore, currently there is no database that includes selenium concentrations in foods in Germany. Nevertheless, it was possible to use data from neighbouring European countries with similar soil selenium concentrations. Moreover, some of the foods consumed in Germany that contribute to selenium intake are imported from other countries, particularly legumes, nuts, and seeds, but to some extent also grains (especially rice, pseudocereals, durum wheat, and oats) [33]. The intake of fruits and vegetables generally does not strongly contribute to selenium intake [4]. Consequently, due to the uncertainty and variability in the selenium content of foods, the selenium intake reported in the present article should be viewed as a rough approximation of true selenium intake. In addition, in the present study, selenium intake from supplements was not available and therefore could not be included. This is a strong limitation, particularly given vegan children in this study, where selenium supplementation was documented for only 12% of the children (while the amounts of selenium in these supplements was not documented).

Nevertheless, the present study had a large sample size within a narrowly defined, understudied age group and appears to be the first to document selenium intake in vegan children. Furthermore, weighed dietary records were used to provide the best possible estimate of dietary intake for children in this age group [34], and the very detailed food diaries and the splitting up of composite foods into their components made it possible to assign a selenium content to almost all foods based on European data (as provided by the EFSA’s food composition database). Finally, the present study expands current knowledge about the selenium intake of young vegetarian and vegan children, and for the first time estimates are now available on how the selenium intake of vegetarian and vegan children (1-3 years old) differs from that of their omnivorous counterparts and to what extent these dietary groups achieve intake levels of selenium that are considered adequate.

## 5. Conclusions

To our knowledge, the present study is the first to report the selenium intake of vegetarian and vegan children. We observed a lower median selenium intake in vegetarian and vegan children compared to omnivorous children, but this was only significant for vegan children. However, on average, the vegetarian and vegan children also achieved the H-AR for selenium. The finding that 52% of vegetarians, 47% of vegans, and 28% of omnivores did not reach the H-AR of 17 µg/day highlights the fact that diets for young children should be well planned. The results also highlight that a long-term excessive intake of Brazil nuts should probably be avoided for young children, as otherwise the H-UL for selenium intake for young children (60 µg/day) may easily be exceeded.

In future studies, the (additional) selenium intake via supplements and the selenium status based on blood parameters, as well as associations between selenium intake and selenium status, should be investigated among vegetarian, vegan, and omnivorous children.

## Figures and Tables

**Figure 1 nutrients-15-00034-f001:**
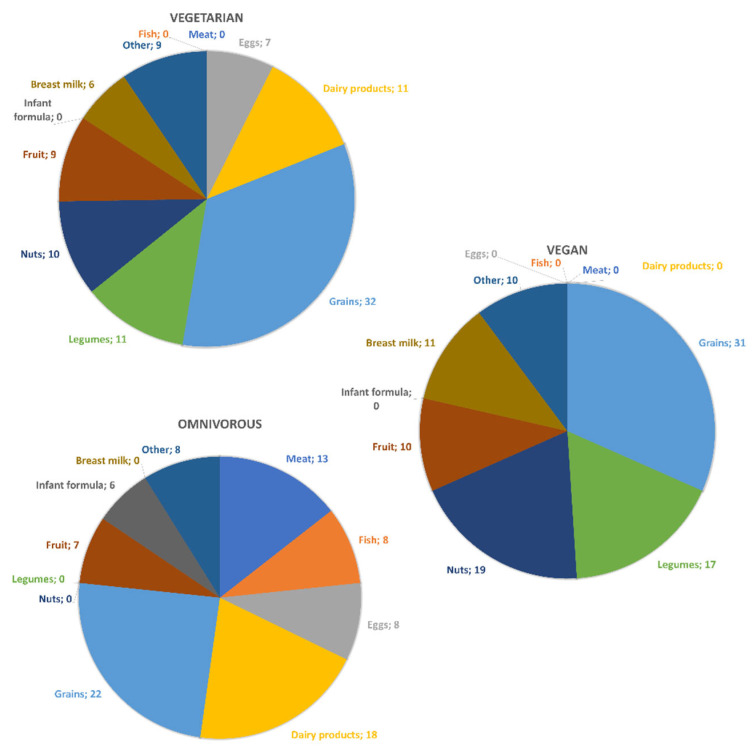
Percentage of selenium intake obtained from different food groups with a contribution of at least 5% to total selenium intake in vegetarian, vegan, and omnivorous 1- to 3-year-olds. 0 denotes 0% or <5%. Food groups which contributed 0–5% of selenium are: vegetarian (meat, fish, animal fats, and plant oils: 0% each; infant formula: 4%; potatoes: 2%; other vegetables: 4%; beverages: 3%), vegan (meat, fish, eggs, dairy, infant formula, and animal fats: 0% each; plant oils: 1%; potatoes: 2%; other vegetables: 4%; beverages: 3%), omnivorous (legumes: 3%; nuts: 3%; breast milk: 2%; animal fats: 0%; plant oils: 1%; potatoes: 2%; other vegetables: 2%; beverages: 3%).

**Table 1 nutrients-15-00034-t001:** Median daily selenium intake among 1- to 3-year-old children by dietary group compared with the H-AR and H-UL.

	VN *n* = 139	VG *n* = 127	OM *n* = 164	H-AR (μg/day)	H-UL (μg/day)	Basic Model *		Final Model	
						*p*	Partial *η^2^*	*p*	Partial *η^2^*
Unadjusted median selenium intake (μg/day) ^a^	19 (13–25) ^1^	17 (14–24)	22 (16–26) ^1^	17	60	0.062	0.013	<0.001	0.058
Unadjusted median selenium intake excluding Brazil nuts (µg/day) ^b^	18 (13–24) ^1^	17 (14–23)	22 (16–26) ^1^	-	-	<0.001	0.044	<0.001	0.047
Unadjusted median selenium intake (µg/1000 kcal/day) ^c^	19 (14–25) ^1^	19 (15–24)	22 (17–26) ^1^	-	-	0.105	0.011	<0.001	0.044

Vegan (VN); vegetarian (VG); omnivorous (OM); harmonized average requirement (H-AR); harmonized tolerable upper intake level (H-UL). Values are given as the median (25th to 75th percentile). * *p*-values and effect sizes were determined using ANCOVA, adjusting for age and sex. ^a^
*p*-values and effect sizes of the final model were determined with ANCOVA, adjusting for age, sex, weight-to-height Z-scores, season, socio-economic status, and total energy intake (*n* = 430). ^b^
*p*-values and effect sizes of the final model were determined with ANCOVA (log[x]-transformed), adjusting for age, sex, weight-to-height Z-scores, season, socio-economic status, and total energy intake (*n* = 430). ^c^
*p*-values and effect sizes of the final model were determined with ANCOVA (log[x]-transformed), adjusting for age, sex, weight-to-height Z-scores, season, and socio-economic status (*n* = 423). ^1^ Values with the same superscript were significantly different in the fully adjusted model (*p* <0.001).

## Data Availability

The data are available from the corresponding author upon reasonable request.

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
