# Peer review of "How Does Selenium Intake Differ among Children (1–3 Years) on Vegetarian, Vegan, and Omnivorous Diets? Results of the VeChi Diet Study"

_nutrients, 2022, doi:10.3390/nu15010034_

Round 1
Reviewer 1 Report
I regard this paper as a great paper on a specific mineral in plant-based vs general diet in very young children. Children between 1 and 5 years are mostly refer to as “young children”; also is this the definition by World Health organization. When the word ‘children’ is used, it is mostly for 5-12 years. Therefore I strongly recommend to mention that in the introduction. When reading this paper for 1-3 years old, I am wondering how the selenium intake will be in older children in plant-based and normal diet. This is also depending on which cut-off or recommended daily intake is used in the world. Mostly recommended daily intake are estimated, so we do not really know if 17 microg/d is really enough for 1-3 years olds.
Line 16. Why mentioning ‘selenium status’ in abstract as that is not part of this paper?
Line 43. Same comment. Why mentioning here selenium status of vegan children, as that is not what is investigated in this paper? I expected some results of status, but later only data of intake are shown.
Line 66. What is the ‘volume of breast per breast milk meal’? I am already many years in infant formula research and this is so depending on age of the child and how many ‘meals/breast milk’ per day, that an average volume cannot be used.
Line 80. ‘intermediate to low soil selenium concentrations’. I hope that you do not only count the selenium levels in consumed plants/vegetables/fruits. There is also selenium in dairy products and other animal-based food. From the text, it is not clear if all dietary selenium is counted in the consumed diets.
Line 92-93. Unbelievable…. If the chosen cohort with only 3-day dietary records did not documented the dietary supplements, then the estimated selenium intake is really a rough estimate. Please indicate this shortcoming in the discussion.
Line 105. What is that harmonized average requirement (%H-AR)? Is there not a recommended daily intake for selenium? I think that the authors should look to an official recommended intake for selenium for this age group.
Line 106. If the authors do not take dietary supplements into account, then looking to tolerable upper level of intake of selenium does not make sense, as for food intake only it is very very hard to reach upper levels of minerals. The dietary supplements are mostly the reason.
I miss the characteristics of the young children. How old were the children exactly? As a diet for 1 year and 2 months old young child is totally different from a diet of an almost 3 year old. How many infants were between 1 and 2 year? How many between 2 and 3 years? They will surely consume different types of food and different amount of food. Where the young children healthy? Living at home with parents? Did the parents prepare the food for the young children? Or were the young children mostly at the day care and so all kind of different food was consumed?
Line 178. Why using the term ‘toddlers’ here? Also in line 199.
Line 179. Of course median selenium intake is the highest in omnivorous group, as they also consume meat, fish and dairy products. Please give this explanation also. It is exactly as expected.
Line 192. It is very strange that these results are written in the discussion part and refer to a figure. These text should be in the results part written.
Line 196. So there is a comparable German study of selenium intake in very young children. Were the methods the same? Did they count the dietary supplements? To which intake did they compare? Did they also conclude that there is no problem with the current selenium intake in this age group?
Line 205-207. No clue where these percentages and amount come from and why it is mentioned and what you want with this 15 microg/d number?
Line 274. What is the dietary reference values for selenium for this age group? It is very normal to compare intake to recommended values, so please do. It is still unclear where they got the 17 microg/d from? And when choosing H-AR, where does the H-AR come from and why they used that?
Author Response
Thank you very much for your helpful feedback. We have added our replies in square brackets below your comments:
I regard this paper as a great paper on a specific mineral in plant-based vs general diet in very young children. Children between 1 and 5 years are mostly refer to as “young children”; also is this the definition by World Health organization. When the word ‘children’ is used, it is mostly for 5-12 years. Therefore I strongly recommend to mention that in the introduction. When reading this paper for 1-3 years old, I am wondering how the selenium intake will be in older children in plant-based and normal diet. This is also depending on which cut-off or recommended daily intake is used in the world. Mostly recommended daily intake are estimated, so we do not really know if 17 microg/d is really enough for 1-3 years olds.
[We have now added “including 1- to 3-year-old children” in the Introduction to clarifiy that the VeChi Diet study included only young children in this age range. As the “(1–3 years)” is also mentioned in the title, we hope that this will avoid confusion for the reader. As mentioned below, we have now also included further reference values (German dietary reference intake: 15 µg/d; EFSA adequate intake: 15 µg/d; United States recommended dietary allowance: 20 µg/d).]
Line 16. Why mentioning ‘selenium status’ in abstract as that is not part of this paper?
[Thank you very much for your comments. We have now deleted the reference to „selenium status“.]
Line 43. Same comment. Why mentioning here selenium status of vegan children, as that is not what is investigated in this paper? I expected some results of status, but later only data of intake are shown.
[We have now deleted the reference to „selenium status“.]
Line 66. What is the ‘volume of breast per breast milk meal’? I am already many years in infant formula research and this is so depending on age of the child and how many ‘meals/breast milk’ per day, that an average volume cannot be used.
[Thank you for pointing out that this is unclear. We have updated the sentence and added „age-specific“ and „as reported previously“. The methodology was the same as the one used in the present study (VeChi Diet study) for the analysis of other nutrients. That is, the age of the children was taken into account. The exact volume of each breast milk meal was not assessed unfortunately. So, it had to be estimated.].
Line 80. ‘intermediate to low soil selenium concentrations’. I hope that you do not only count the selenium levels in consumed plants/vegetables/fruits. There is also selenium in dairy products and other animal-based food. From the text, it is not clear if all dietary selenium is counted in the consumed diets.
[As described by Combs 2011 (doi: 10.1079/BJN2000280), soil selenium levels are a factor that influence the variability of selenium intake and status in different regions of the world. For the lacto-ovo-vegetarian and the omnivorous children, animal-source foods expectedly also contributed to selenium intake, which can be seen in Figure 1. In line 93 (perviously line 80), we are referring to a comparison of four European countries with selenium soil concentrations that appear to be similar to Germany. This seems reasonable because using data from countries with higher soil selenium levels would lead to overestimating selenium intake from plant foods, and grains contribute a large percentage of selenium intake (which can be seen in Figure 1). Therefore, we have not made any changes in this section.]
Line 92-93. Unbelievable…. If the chosen cohort with only 3-day dietary records did not documented the dietary supplements, then the estimated selenium intake is really a rough estimate. Please indicate this shortcoming in the discussion.
[We agree with the reviewer, and this limitation is mentioned in the Discussion. We have now added: „This is a strong limitation, especially regarding the selenium intake of the vegan children in this study, for 12% of whom a selenium supplementation was documented..“]
Line 105. What is that harmonized average requirement (%H-AR)? Is there not a recommended daily intake for selenium? I think that the authors should look to an official recommended intake for selenium for this age group.
[We agree that recommended daily intakes are more commonly used. However, recently, a globally applicable reference value has been proposed: The H-AR is „an approach for harmonizing the NRVs [nutrient intake reference values] for ARs [average requirement] (here termed “H-ARs”) and ULs (“H-ULs”) that can be applied on a global scale to assessing intakes across populations“ (doi: 10.1093/advances/nmz096, reference 13 in the present manuscript). However, based on your suggestion we have now added the following to the manuscript: “(the adequate intake for selenium given by EFSA for 1- to 3-year-olds). It should be taken into account that in the German-speaking countries (Germany, Austria, and Switzerland), the dietary reference value for selenium for children 1- to 4-year-old children is 15 µg/d, while in the United States, the dietary reference value (adequate intake) for 1- to 3-year-olds is 20 µg/d. Thus, the majority of the children in the present study did not reach the selenium intake recommended in the United States.” (lines 243-249)]
Line 106. If the authors do not take dietary supplements into account, then looking to tolerable upper level of intake of selenium does not make sense, as for food intake only it is very very hard to reach upper levels of minerals. The dietary supplements are mostly the reason.
[We agree that not having the amounts of selenium consumed via supplements is a strong limitation. However, the data presented does suggest that the prevalence of selenium supplementation was very low among lacto-ovo-vegetarians (2%) and omnivores (1%). The data presented also show that „Nine vegan and two vegetarian children exceeded the H-UL of 60 µg/d (by 110–775%)“, i.e. exceeding the H-UL was a possibility even though supplements were not taken into account.]
I miss the characteristics of the young children. How old were the children exactly? As a diet for 1 year and 2 months old young child is totally different from a diet of an almost 3 year old. How many infants were between 1 and 2 year? How many between 2 and 3 years? They will surely consume different types of food and different amount of food. Where the young children healthy? Living at home with parents? Did the parents prepare the food for the young children? Or were the young children mostly at the day care and so all kind of different food was consumed?
[We have added this to the Results section: “53% of the vegetarian, 58% of the vegan, and 52% of the omnivorous children were <2 years old. 29% of the vegetarian, 27% of the vegan, and 27% of the omnivorous children were ≥2 and <3 years old, while 18% of the vegetarian, 14% of the vegan, and 21% of the omnivorous children were already 3 years old. More detailed characteristics of the participants have been published previously [1].” (lines 143-147) The children were all healthy. We have added “healthy” and “living in Germany” in the Methods section (lines 66-67). The children lived at home with their parents, and the parents prepared most of their food. Some children also attended day care. It can be assumed practically all the food for the vegan children was prepared at home.]
Line 178. Why using the term ‘toddlers’ here? Also in line 199.
[We have now changed all mentions of “toddlers” to “children” or “young children” or “children (1 to 3 years old)” to avoid confusion.]
Line 179. Of course median selenium intake is the highest in omnivorous group, as they also consume meat, fish and dairy products. Please give this explanation also. It is exactly as expected.
[This has been mentioned in the Discussion section: “A possible explanation for a lower selenium intake in vegetarians may simply be that they do not consume meat or fish and (like in the present study) also fewer dairy products compared to omnivores (data not shown), whereas omnivores obtained ~50% of their dietary selenium from animal-source foods (meat, dairy products, eggs, and fish), which tend to be rich in selenium (Figure 1)”. We have now added the following sentence: “Thus, the expectation that selenium intake would be higher in omnivorous children, due to their intake of animal-source foods, was confirmed.”]
Line 192. It is very strange that these results are written in the discussion part and refer to a figure. These text should be in the results part written.
[We have added the following sentences to the Results section: “It should be noted that 7 of the 139 vegan children (5%) consumed some lacto-ovo-vegetarian food when in day care, and 3 of the 127 vegetarian children (2%) also consumed some non-vegetarian food when in day care. However, the contribution of these animal-source foods to the respective selenium intakes of vegan and vegetarian children were minimal (<0.5%; Figure 1).” (lines 179-183)]
Line 196. So there is a comparable German study of selenium intake in very young children. Were the methods the same? Did they count the dietary supplements? To which intake did they compare? Did they also conclude that there is no problem with the current selenium intake in this age group?
[We have cited this data from the “EFSA. Scientific Opinion on Dietary Reference Values for Selenium” (https://www.efsa.europa.eu/sites/default/files/consultation/140715.pdf). To the best of our knowlegde the results of the VELS study (2003) regarding selenium have not been published elsewhere.
The VELS study was conducted for the German Federal Institute of Risk Assessment (BfR). The VELS study study’s aim was to assess the food intake of infants and young children in order to estimate the risk of acute toxicity from contaminants in food originating from pesticides (German title of the study: Verzehrsstudie zur Ermittlung der Lebensmittelaufnahme von Säuglingen und Kleinkindern für die Abschätzung eines akuten Toxizitätsrisikos durch Rückstände von Pflanzenschutzmitteln). Thus, the purpose of the VELS study was not to assess the adequacy of dietary selenium intake. Selenium intake was later estimated by EFSA based on data from the VELS study.
The methods were the same in that dietary intake was also assessed with a 3-day-protocol (at two timepoints, 3 to 6 months apart), As in the VeChi study dietary intake from supplements was not included in the VELS study. In the VELS study, there was no comparison group. Data regarding selenium intake is not contained in the original VELS study report, and, consequently, there was no assessment of the adequacy of selenium intake.]
Line 205-207. No clue where these percentages and amount come from and why it is mentioned and what you want with this 15 microg/d number?
[Thank you for pointing out that it was unclear what the 15 µg/d value refers to. The percentage values mentioned in the Discussion (that you are referring to) are mentioned in the Results section (“39% of vegetarians, 36% of vegans, and 16% of omnivores had a selenium intake of <15 µg/d”, lines 157-159). We have now added: “(15 µg/d is the adequate intake for selenium for 1- to 3-year-olds given by EFSA)”, lines 159-160]
Line 274. What is the dietary reference values for selenium for this age group? It is very normal to compare intake to recommended values, so please do. It is still unclear where they got the 17 microg/d from? And when choosing H-AR, where does the H-AR come from and why they used that?
[Thank you for pointing out this lack of clarity. Different dietary reference intakes for selenium are given for this age group, which has now been added to the manuscript (lines 244-249). In the German-speaking countries (Germany, Austria, and Switzerland), the dietary reference value for selenium for 1- to 4-year-old children is 15 µg/d. The adequate intake for selenium given by EFSA for 1- to 3-year-olds is also 15 µg/d. However, in the United States, the dietary reference value for selenium for 1- to 3-year-olds is 20 µg/d.
To harmonize these different values, the concept of H-AR has been proposed by Allen et al. 2020 (doi:10.1093/advances/nmz096), wich is 17 µg/d and is based on (equal to) the Estimated Average Requirement (EAR) given by the Institute of Medicine. We have now added the following sentence: “The H-AR for selenium for 1- to 3-year-olds is 17 µg/d [13] and is based on the Estimated Average Requirement (EAR) given by the Institute of Medicine (IOM, United States) in the year 2000” (lines 122-124). In our mansucript we have compared the selenium intake to the H-AR (17 µg/d) as well as the dietary reference intakes recommended in Germany, Austria, and Switzerland (these countries jointly provide recommendations) as well as by the European Food Safety Authority (EFSA; 15 mg/d). We have now, however, also mentioned the higher RDA (20 µg/d) in the United States (line 247).]
Reviewer 2 Report
Section 1: Abstract:
Q1. The expressions need to be further improved and simplified.
Section 2: Introduction:
Q2. The last paragraph of Introduction section should present the research contents, significance, innovation and prospects of this study, which is not recommended to predict/reveal the results of the study.
Section 3: Materials and Methods:
Q3. Whether the specific positions of meat and fish were were included in the study, which would significantly affect the selenium content and dietary preferences.
Section 4: Results:
Q4. Why were these data not presented as mean values ± standard deviation? What is the relative merits?
Q5. Does the unadjusted median selenium intake of 19, 17, 22 (mg/d) present any significant variation, as these values are all above the minimum intake recommendations and below the dangerous dose for adverse effects.
Section 5: Discussion:
Q6. The strengths related expressions were suggested to be further improved in line 249~274.
Author Response
Thank you very much for your helpful comments. Below each of your comments we have written our reply in square brackets:
Section 1: Abstract:
Q1. The expressions need to be further improved and simplified.
[We have rewritten the Abstract to make the sentences clearer.]
Section 2: Introduction:
Q2. The last paragraph of Introduction section should present the research contents, significance, innovation and prospects of this study, which is not recommended to predict/reveal the results of the study.
[We have changed this paragraph (lines 48-60) and have moved the study’s hypothesis to the Materials and Methods section (2.1. Study Design and Participants; lines 70-72).]
Section 3: Materials and Methods:
Q3. Whether the specific positions of meat and fish were were included in the study, which would significantly affect the selenium content and dietary preferences.
[We have added this sentence: “If any of the children categorized as vegetarian or vegan children also consumed small amounts of animal-source foods, selenium intake of these animal-source foods was also included.” (lines 108-110)]
Section 4: Results:
Q4. Why were these data not presented as mean values ± standard deviation? What is the relative merits?
[As these variables were not normally distributed, we chose to present the data as median (25th and 75th percentile) rather than mean ± standard deviation.]
Q5. Does the unadjusted median selenium intake of 19, 17, 22 (mg/d) present any significant variation, as these values are all above the minimum intake recommendations and below the dangerous dose for adverse effects.
[The unadjusted median with the 25th and 75th percentiles of selenium intake can be seen in Table 1, i.e. the 25th percentile of selenium intake of vegan, vegetarian, and omnivorous children was 13, 14, and 16 µg/d.]
Section 5: Discussion:
Q6. The strengths related expressions were suggested to be further improved in line 249~274.
[We have rewritten this section (lines 296-325).]